# Attachment beyond the screen: The influences of demographic factors and parasocial relationships on social media use in Qatar

**Ruining Jin** [iD] [1,2]*, **Tam-Tri Le** [3]

**1** Institute of Higher Education, Beijing University of Technology, Beijing, China, **2** Capital Engineering Education Development Research Base, Beijing, China, **3** Independent researcher, Ho Chi Minh City, Vietnam

* ruiningjin@bjut.edu.cn

## Abstract

### Background

Most studies on social media usage and parasocial relationships (PSRs) have been conducted in WEIRD (Western, Educated, Industrialized, Rich, and Democratic) societies, potentially overlooking the unique cultural, social, and economic factors present in non-WEIRD contexts. Examining these phenomena in a non-WEIRD setting is essential for a comprehensive understanding of social media's global impact.

### Methods

Secondary data from 574 participants in Qatar who followed Instagram influencers were analyzed using Bayesian analyses aided by Markov Chain Monte Carlo (MCMC) algorithms to examine the relationships between social media usage time, PSRs, and demographic factors.

### Findings

The analysis results show that, regarding linear effects, a stronger parasocial relationship with Instagram influencer(s) is associated with higher daily social media usage time. Meanwhile, being male, being older, and having higher incomes all have negative associations with daily social media usage time. When parasocial relationships and the three demographic factors are seen in their interactions, negative associations with social media usage were also found in a similar pattern. To elaborate, among those with high parasocial relationship degrees, females, young people, and poor people tend to use social media for more hours each day.

**Data availability statement:** The data is available online at: https://osf.io/jp9e5/?view_only=e4abb1409801403c81cad3c4f7dbbf40.

**Funding:** The author(s) received no specific funding for this work.

**Competing interests:** The authors have declared that no competing interests exist.

## Conclusions

This study highlights that demographic factors such as gender, age, and income in their interactions with parasocial relationships are associated with social media usage time within the non-WEIRD social context of Qatar. The findings underscore the necessity of considering the specific local cultural settings when studying social media behaviors.

---

## 1. Introduction

Social media platforms have become a cornerstone of modern communication, enabling users to connect, share, and consume content with others on a global scale. In 2023, it was estimated that there were over 4.9 billion social media users around the globe, which accounted for approximately 61.2% of the global population [1]. Among these social media platforms, Facebook leads the popularity globally with around 2.9 billion monthly active users, followed by YouTube and WhatsApp, each with over 2 billion users [2]. Famous content creators and influencers such as Kim Kardashian, PewDiePie, and Charli D'Amelio have hundreds of millions of followers, exerting far-reaching impacts in consuming, gaming, pop cultures, or even political views among the younger generation [3–5].

### 1.1. Excessive social media use and its outcomes

While social media offers numerous benefits, it is crucial to address the challenges that have emerged alongside its growth. One of the significant issues that emerged after the explosive growth of social media is the excessive amount of time users spend on these platforms. This aspect has gained attention widely in the Western-educated, Industrialized, Rich, and Democratic (WEIRD) context. In the United States, people average 37 min to 2 hours and 16 min per day of social media usage time [6]. In Germany, before COVID-19, people averaged 2.74 hours of social media usage time, and the COVID-19 restriction extended the average usage time to 3.74 hours [7]. Similarly, in Portugal, a study stated that Portuguese users average 2.5 hours of daily social media usage time [8]. Long-time use of social media has been found to be associated with several problems, including decreased productivity, and sleep disturbances [9,10]. Long time social media use can also bring about depression and other mental health disorders among vulnerable groups [11]. Furthermore, increased time spent on social media was suggested to be related to other conduct problems such as strong alcohol use among adolescents [12].

### 1.2. Social media use and associated demographic factors

Understanding the impact of excessive social media use requires examining the demographic factors that influence usage patterns. Demographic variables such as age, gender, and income significantly impact social media use time, shaping how different groups engage with digital platforms. Younger users, particularly adolescents, and young adults, are more likely to spend extensive time on social media [13–16].

On the other hand, females are more likely than men to use social media to strengthen their social ties [14–16]. Lastly, income is also reported to be associated with social media addiction, as one study concluded that lower-income people tend to be more likely to suffer from social media addiction [14].

## 1.3. PSRs and social media use

In addition to demographic factors, the nature of relationships formed on social media, particularly PSRs, plays a critical role in user behavior and psychological outcomes. PSRs refer to the one-way attachment that individuals form with media figures, such as influencers and celebrities [17]. PSRs can offer comfort to individuals and help form certain identities among fans, which could be helpful for those experiencing loneliness or social anxiety [18]. However, they also pose significant psychological risks. First of all, individuals with low self-esteem may develop unhealthy attachment patterns, which would be detrimental to their mental health conditions [19]. Second, PSRs can also be linked to issues such as social isolation and emotional distress [20]. Furthermore, PSRs might lead to unrealistic expectations of relationships, which would bring about difficulties in forming and maintaining healthy interpersonal connections [11,21].

Given the influence of PSRs on user behavior, it is important to understand how these relationships interact with social media usage patterns. However, only a few studies probed the relationship between PSRs and social media use time. One study suggested that individuals with a high amount of social media use is associated with a high level of parasocial interaction toward Korean Pop Stars [22]. Another study also corroborated that social media usage time is a significant predictor of PSRs, particularly parasocial friendships and emotional connections with media personalities [23].

According to the Uses and Gratifications Theory (UGT), individuals would consume media content to satisfy their specific socio-psychological needs, such as the need for social connection, entertainment, or self-validation [24,25]. In the case of social media users, susceptible demographic groups may desire to gratify various social identity-based needs through interactions with influencers and celebrities, which might be associated with extensive social media usage time and PSRs.

## 1.6. Current study

However, findings from WEIRD contexts may not directly generalize to non-WEIRD populations, because various cultural, socioeconomic, and social norms might significantly influence the social media use pattern, and the formation and impact of PSRs. Currently, studies on social media use in non-WERID contexts offered complicated findings. For example, Chinese netizens average about 2 hours per day social media use time [26]. One study on Chinese adolescents suggested that social media use time alone is not associated with increased anxiety toward their body image. However, when seeking attention through the use of social media, the more social media use time Chinese adolescents, the greater they would suffer from social appearance anxiety [27]. On the other hand, India has more than 600 million social media users [28], one studies on Indian adolescents offered similar findings, suggesting that long time use of social media is associated with increased stress, anxiety, and depression [29]. However, few studies have been conducted on non-WEIRD populations in the Middle East. One cross-cultural study examined the mindful use of social media among Iranian and American users, finding that Iranian participants reported higher levels of mindful social media use, which were associated with lower symptoms of social media addiction [30]. Furthermore, one recent study investigated the role of PSRs with favorite food influencers among Iranian social media users. Their study found that stronger PSRs with food influencers were associated with higher levels of eating disorder symptoms, food addiction, and grazing behaviors [31]. However, the aforementioned studies primarily examined social media use and PSRs within Iran, leaving a significant gap in research on non-WEIRD populations in the broader Middle Eastern context. Qatar's unique context offers a distinctive environment where rapid economic development and high internet penetration coexist with deeply rooted cultural and traditional norms. This blend creates a unique dynamic in how individuals engage with social media and form PSRs with influencers. Furthermore, due to the impact of COVID-19, nationwide lockdowns and social-distancing measures accelerated the adoption of digital

platforms and heightened reliance on Instagram for entertainment, communication, and shopping, pushing more users to interact with influencers and branded content online [1–3]. In fact, recent estimates suggest that Instagram's penetration in Qatar grew by over 10% between 2020 and 2022, and influencers took on a more prominent role in disseminating public information and shaping consumer habits during the pandemic [4,5]. As of January 2024, Instagram users have been reported to reach 1.7 million in 2024 (up from 1.1 million in 2023) with 35.1% female users and 64.9% male users [32,33].

As a result of the limited research on non-WEIRD Middle Eastern populations and the rapid, COVID-19–induced shifts in Instagram adoption and influencer engagement, understanding these dynamics in Qatar's digital spaces not only contributes to the social media development in Qatar but also offers novel insights into similar regions of quickly developing or emerging economies, where technological development and modernization meet with conservative values and ideologies. To this end, the present study has the following research questions (RQs):

RQ1: In linear relationships, how may social media usage time be associated with PSR, gender, age, and income?

RQ2: How may social media usage time be associated with the interacted influences from PSR and the above demographic factors?

## 2. Methodology

### 2.1. Materials and variables

We use secondary data from the data article "Instagram Influencers Attributes and Parasocial Relationship: A dataset from Qatar" [34]. Research ethics approval for the data collection by [34] was obtained from the Qatar University Review Board (number QU-IRB 1195-E/19). [34] declared that participation was entirely voluntary, all respondents were systematically informed about the study's content and objectives before participation, and all respondents gave informed consent to participate. The data collection happened in 2020, from January 29th to February 16th.

The dataset contains survey information from 574 participants living in Qatar who followed Instagram influencers. Participants were required to follow at least one Instagram influencer from the following areas of expertise: Fashion (N = 26, 4.5%), Traveling (N = 30, 5.2%), Beauty Products (N = 16, 2.8%), Food and Beverages (N = 44, 7.7%), Others (N = 272, 47.4%), and Multiple (N = 186, 32.4%). Among these participants, 38.3% of the participants were males (N = 220) and 61.7% were females (N = 354). The participants were divided into age groups: 18–24 (N = 375, 65.3%), 25–34 (N = 142, 24.7%), 35–44 (N = 45, 7.8%), 45–54 (N = 10, 1.7%), 55–64 (N = 2, 0.3%). Regarding income (measured in Qatari Riyals), there were 6 groups: < 50,000 (N = 354, 61.7%), 50,000–150,000 (N = 116, 20.2%), 150,000–250,000 (N = 48, 8.4%), 250,000–350,000 (N = 29, 5.1%), 350,000–450,000 (N = 10, 1.7%), > 450,000 (N = 17, 3%). About half of the participants spent more than 5 hours every day on social media. There were no participants who did not use social media on a daily basis. More details on the data collection process and basic statistics are available openly online in the original data article [34].

The degree of PSR was measured using the adapted scale based on the study by [35], which included 6 items. Answers were scored on a 5-point Likert scale ranging from "1" being "strongly disagree" to "5" being "strongly agree". The Cronbach's alpha value for the PSR scale in the dataset is 0.843 [34].

The variables used for analysis in this study are presented in Table 1.

### 2.2. Analysis procedure

In the present study, two analytical models for regression were constructed. Model 1 examines multiple linear relationships where *time* is the outcome variable. Model 1 is as follows.

$$\mu_i = \beta_0 + \beta_{parasocial} * parasocial_i + \beta_{gender} * gender_i + \beta_{age} * age_i + \beta_{income} * income_i \qquad (1)$$

**Table 1. Variable description.**

| Variable | Description | Value |
|---|---|---|
| time | The participant's number of hours spent on social media every day | 1 is none |
| | | 2 is 1–2 hours |
| | | 3 is 3–4 hours |
| | | 4 is 5 hours or more |
| parasocial | The participant's average score on the PSR scale | Ranging from 1 to 5 |
| gender | The participant's self-reported gender | 0 is female |
| | | 1 is male |
| age | The participant's age group | 1 is 18–24 years old |
| | | 2 is 25–34 years old |
| | | 3 is 35–44 years old |
| | | 4 is 45–54 years old |
| | | 5 is 55–64 years old |
| income | The participant's annual income in Qatari Riyals | 1 is less than 50,000 |
| | | 2 is 50,000–150,000 |
| | | 3 is 150,000–250,000 |
| | | 4 is 250,000–350,000 |
| | | 5 is 350,000–450,000 |
| | | 6 is above 450,000 |

The posterior distributions of *time* are in the form of normal distribution where $\mu_i$ is the mean value of participant $i$'s number of hours spent on social media every day. $parasocial_i$ is participant $i$'s degree of PSR. $gender_i$ is participant $i$'s gender. $age_i$ is the age group that participant $i$ belonged to. $income_i$ is the annual income group that participant $i$ belonged to. Model 1 has an intercept $\beta_0$ and coefficients $\beta_{parasocial}$, $\beta_{gender}$, $\beta_{age}$, and $\beta_{income}$.

Model 2 examines the effects of multiple interactions between *parasocial* and other independent variables toward the outcome *time*. The two models are separated following the principle of parsimonious model construction, which helps increase the predictive power of the inference [36]. Model 2 is as follows.

$$\mu_i = \beta_0 + \beta_{parasocial*gender} * parasocial_i * gender_i + \beta_{parasocial*age} * parasocial_i * age_i + \beta_{parasocial*income} * parasocial_i * income_i \quad (2)$$

Model 2 has an intercept $\beta_0$ and coefficients $\beta_{parasocial*gender}$, $\beta_{parasocial*age}$, and $\beta_{parasocial*income}$.

For statistical analysis, we used Bayesian analysis aided by Markov Chain Monte Carlo (MCMC) algorithms. The analysis procedure and result presentation followed the protocol of MCMC-aided Bayesian analytics for social sciences and psychological research [36]. The dataset used in the present study has a sample size of 574 participants. While this can be considered an acceptable sample size for the measured media-use-related parameters [34], the high skewness in demographic factors (due to the nature of digital social media use) can negatively affect inference accuracy because of the low data points available in some categories. For example, because of the higher proportion of women (61.7% vs. 38.3% men), traditional frequentist analyses can yield less stable parameter estimates under such imbalances or low data counts in specific subgroups. By contrast, a Bayesian framework aided by MCMC simulations allows for more flexible handling of skewed data, as it models parameters as probability distributions rather than fixed values. This approach "borrows strength" from more populated subgroups while explicitly accounting for uncertainty in underrepresented categories. Furthermore, the Bayesian approach treats all parameters probabilistically, and results are interpreted based on the highest probability of occurrence on credible ranges, which helps provide flexible interpretation and high predictive power [37–40].

Analytical models were checked for goodness-of-fit using Pareto-smoothed importance sampling leave-one-out (PSIS-LOO) diagnostics [41,42] to examine if simulated data fit well with the original data. Through the diagnosis run in R, if *k* values are all below 0.5, the model has healthy goodness-of-fit. *k* values above the threshold of 0.7 would indicate problematic observations that can affect the inference. Markov properties in the MCMC processes were checked using statistical indicators including the effective sample size (*n_eff*) and the Gelman-Rubin shrink factor (*Rhat*). *n_eff* values over 1000 are deemed sufficient for reliable inference [43], and *Rhat* values equaling 1 indicate good Markov chain convergence [44,45]. Convergence was also diagnosed using trace plots, Gelman-Rubin-Brooks plots, and autocorrelation plots. The analysis was conducted using the *bayesvl* package in R [46], using uninformative priors to minimize subjective influences. The MCMC setup was 5000 iterations (including 2000 warm-up iterations) and 4 chains.

## 3. Result

### 3.1. Model 1

The PSIS diagnostic result for Model 1 (Fig 1) shows that all *k* values are lower than 0.5, and there are no problematic observations that may influence the inference. The diagnosis indicates that Model 1 has a healthy goodness-of-fit.

The effective sample size (all *n_eff* values greater than 1000) and Gelman-Rubin shrink factor (all *Rhat* values equal 1) show that the Markov chains are well-converged for Model 1 (see Table 2).

The colored lines represent the Markov chains in Model 1's trace plots (Fig 2). In each plot, the chains fluctuate around a central equilibrium after the warmup period (from 2,000[th] iteration), suggesting good convergence. Additionally, the Gelman-Rubin-Brooks plots show that *Rhat* values dropped to 1 during the warm-up period (Fig A1, Appendix). The autocorrelation plots show that problematic autocorrelation among simulated data points within the MCMC processes was quickly eliminated (Fig A2, Appendix).

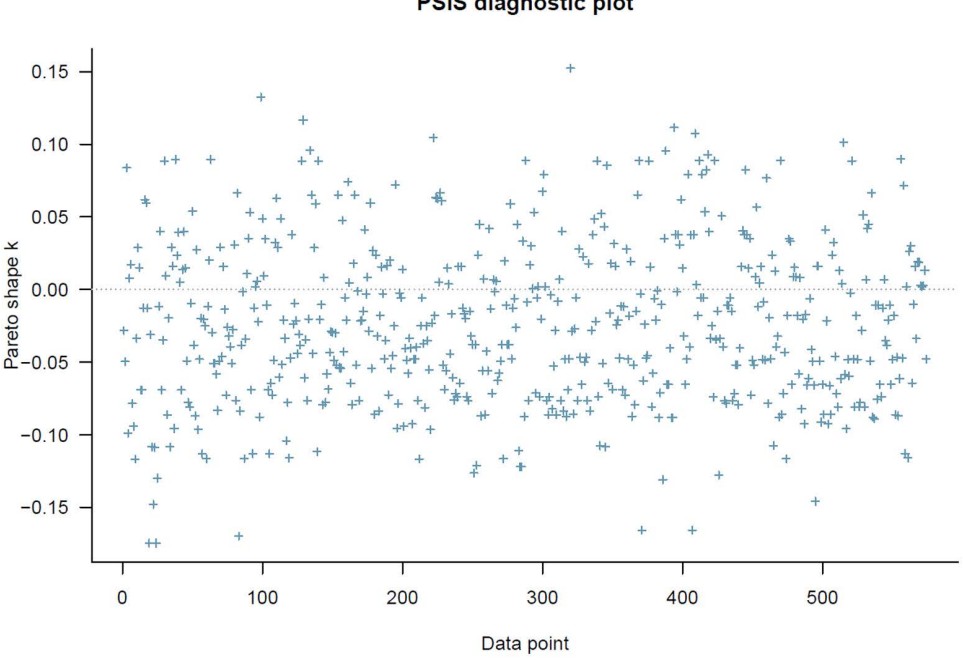

**Fig 1. Model 1's PSIS diagnostic plot.**

**Table 2. Model 1's simulated posteriors.**

| Parameters | Mean (M) | Standard deviation (S) | n_eff | Rhat |
|---|---|---|---|---|
| Constant | 3.59 | 0.13 | 7148 | 1 |
| *parasocial* | 0.05 | 0.03 | 8225 | 1 |
| *gender* | −0.13 | 0.06 | 11520 | 1 |
| *age* | −0.18 | 0.04 | 9348 | 1 |
| *income* | −0.05 | 0.02 | 10110 | 1 |

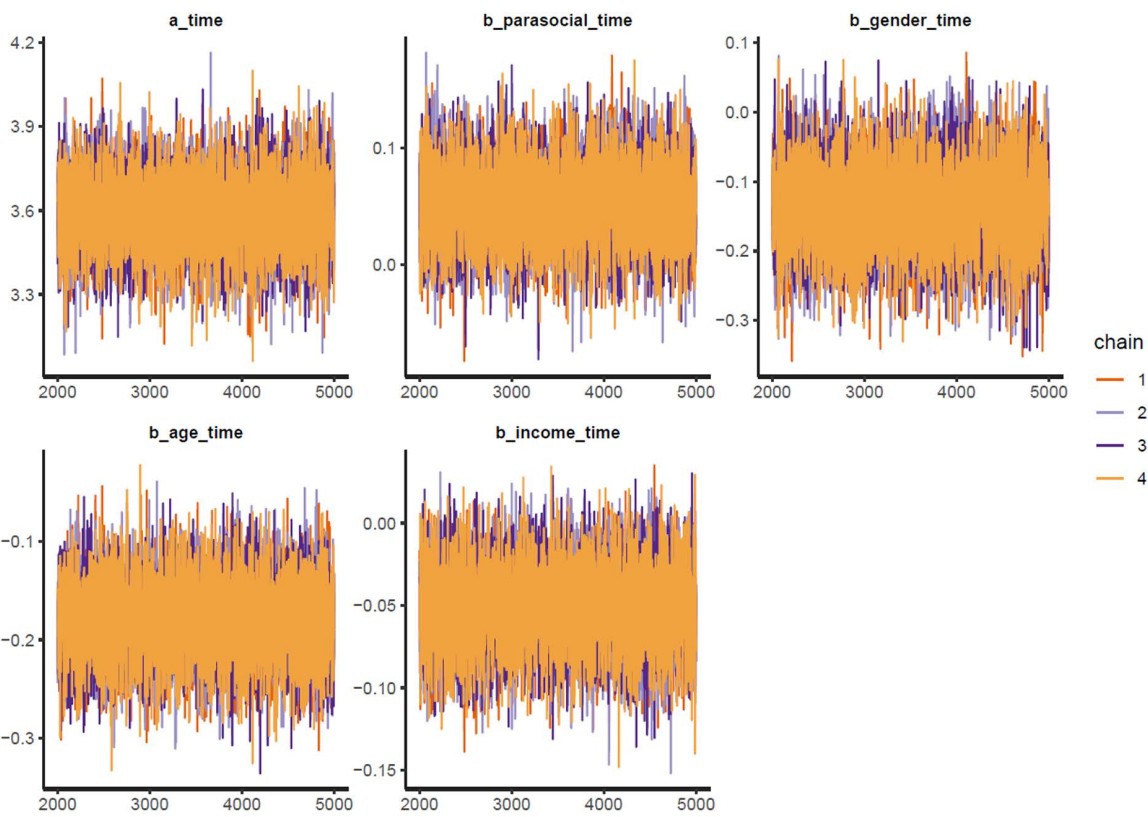

**Fig 2. Model 1's trace plots.**

Estimated posterior coefficients (see Table 2) show that *parasocial* is positively associated with *time* ($M_{parasocial}$ = 0.05 and $S_{parasocial}$ = 0.03). *gender*, *age*, and *income* are all negatively associated with time ($M_{gender}$ = −0.13 and $S_{gender}$ = 0.06, $M_{age}$ = −0.18 and $S_{age}$ = 0.04, $M_{income}$ = −0.05 and $S_{income}$ = 0.02). The effects have good reliability, since the posterior distributions of *parasocial* lie almost completely on the positive side, whereas the posterior distributions of *gender*, *age*, and *income* lie almost completely on the negative side (see Fig 3).

### 3.2. Model 2

The PSIS diagnostic result for Model 2 (Fig 4) also shows that all *k* values are lower than 0.5, indicating no problematic observations.

The values of effective sample size and Gelman-Rubin shrink factor are also healthy for Model 2. As shown in Table 3, all *n_eff* values are greater than 1000, and all *Rhat* values equal 1.

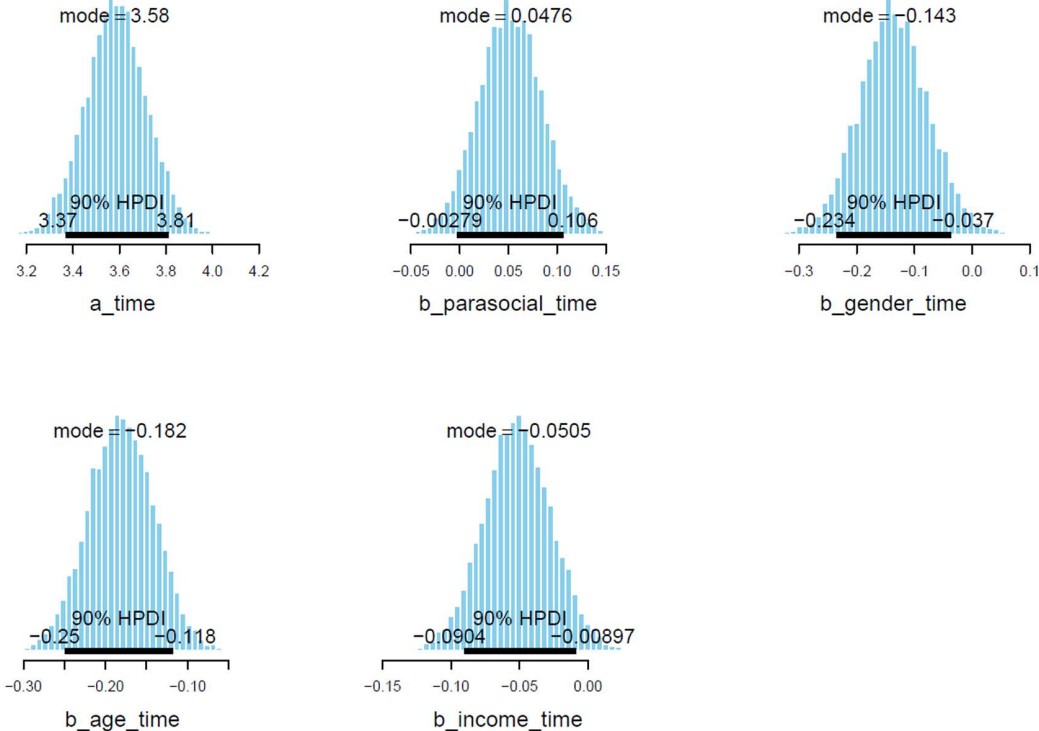

**Fig 3. Model 1's posterior distributions within 90% of Highest Posterior Density Intervals.**

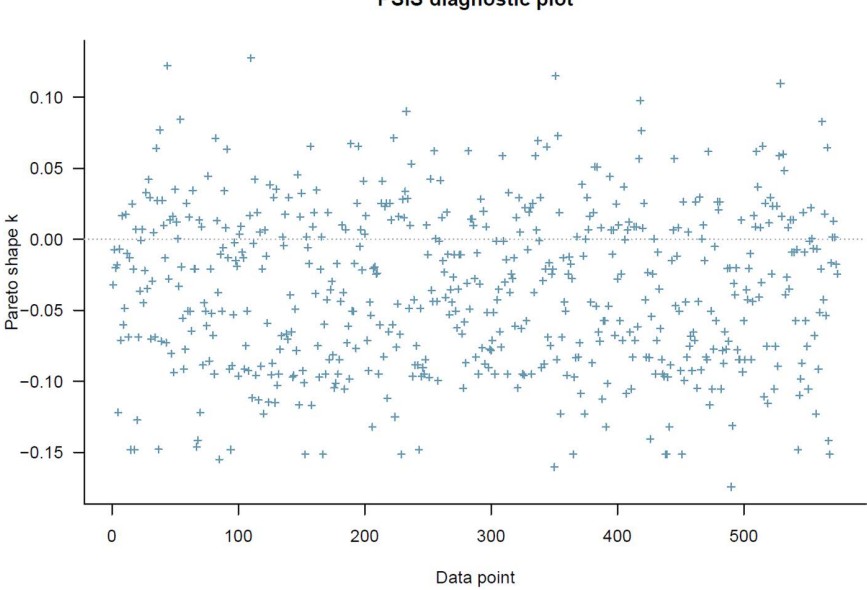

**Fig 4. Model 2's PSIS diagnostic plot.**

The trace plots (Fig 5), Gelman-Rubin-Brooks plots (Fig A3, Appendix), and autocorrelation plots (Fig A4, Appendix) all indicate that Model 2 achieved good Markov properties.

Estimated posterior coefficients (see Table 3) show that all three effects of the independent variable interactions are negative toward *time* ($M_{parasocial\_gender}$ = −0.02 and $S_{parasocial\_gender}$ = 0.02, $M_{parasocial\_age}$ = −0.03 and $S_{parasocial\_age}$ = 0.01, $M_{parasocial\_income}$ = −0.01 and $S_{parasocial\_income}$ = 0.01). The effects have moderate reliability, since the posterior distributions of all three parameters lie mostly on the negative side (see Fig 6).

## 4. Discussion

The analysis results show that, regarding linear effects, a higher PSR with one or multiple favorite Instagram influencers is associated with higher daily social media usage time. Meanwhile, being male, being older, and having higher incomes all have negative associations with daily social media usage time. When PSRs and the three demographic factors are seen in their interactions, negative associations with social media usage were also found in a similar pattern. To elaborate, among those with high PSR degrees with their favorite Instagram influencer(s), females, young people, and poor people tend to use social media for more hours each day.

The finding that a higher PSR degree with Instagram influencer(s) is associated with higher daily social media usage time is in alignment with prior studies [22,23]. Such a finding can be interpreted through a two-way influence. Firstly, when users have a strong PSR with influencers, they are naturally more inclined to engage with the influencers' content (viewing, commenting,

**Table 3. Model 2's simulated posteriors.**

| Parameters | Mean (M) | Standard deviation (S) | n_eff | Rhat |
|---|---|---|---|---|
| Constant | 3.56 | 0.07 | 8743 | 1 |
| *parasocial_gender* | −0.02 | 0.02 | 11219 | 1 |
| *parasocial_age* | −0.03 | 0.01 | 9925 | 1 |
| *parasocial_income* | −0.01 | 0.01 | 12210 | 1 |

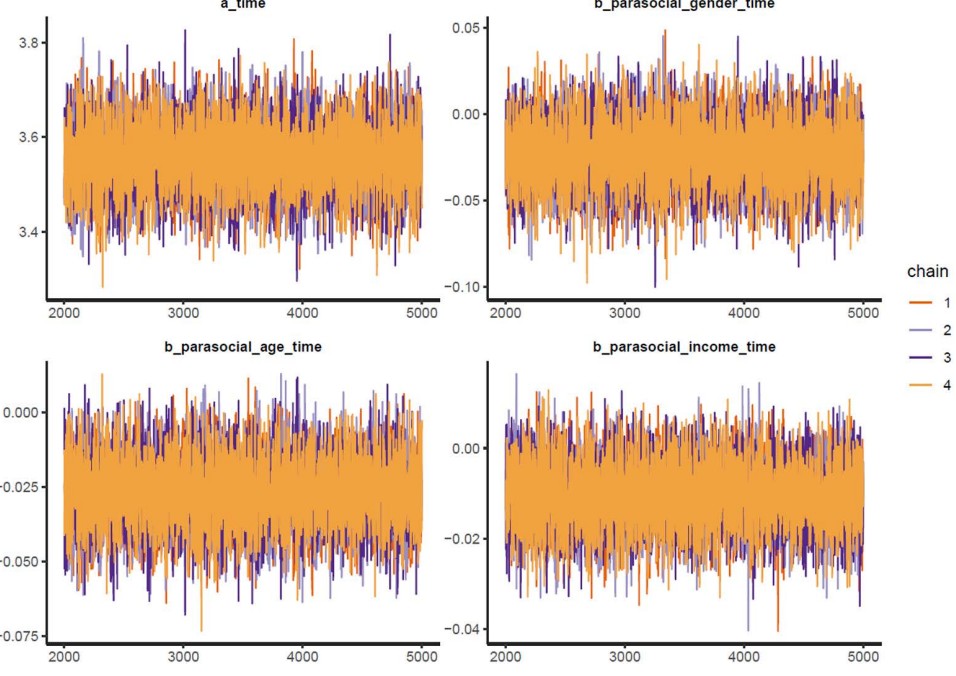

**Fig 5. Model 2's trace plots.**

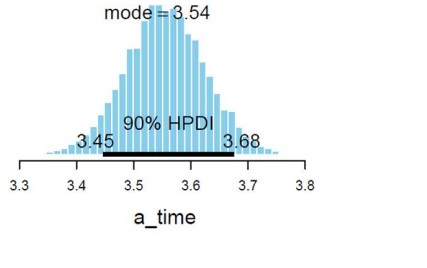
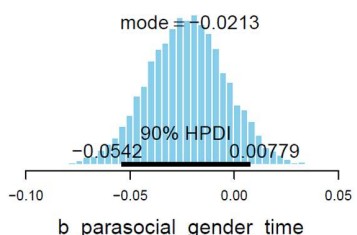
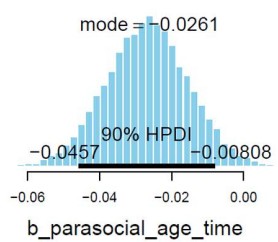
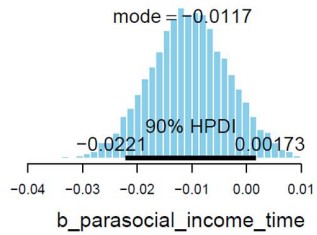

**Fig 6. Model 2's posterior distributions within 90% of Highest Posterior Density Intervals.**

liking, sharing, etc.), which directs the social media platforms to automatically refine the provision of their desired content through algorithms. Thus, the platforms deliver more tailored content that aligns with the user's interests, particularly content related to the influencers they follow [47,48]. This personalized content loop would be associated with increased social media usage, as users are drawn into a continuous cycle of engagement [27]. Reversely, as users spend more time on social media, there is a higher chance for them to come across influencers' content that is engaging and stimulating, which helps form or reinforce the perceived connections between followers and influencers [49]. The felt attachment due to various psychological factors/reasons, as suggested in the UGT [24,25], can further strengthen a PSR through this reciprocal loop of social media content engagement.

The study findings that being male, being older, and having higher incomes have a negative association with daily social media usage time are consistent with earlier studies [14–16,50–52]. Here, we can take into account the specific cultural perspective of the studied population. One possible aspect is that when males in Qatar use social media, they also consider the socially expected masculine characteristics, as well as traditional masculine roles in the family and society, such as showcasing socioeconomic status [53], or highlighting the provider-income maker identity in the family [54,55]. In this sense, long social media usage time might be perceived as excessive leisure or gossiping, undermining male users' ideal image (both self-image and social image). Thus, males' social media usage time is negatively associated with their gender identity.

On the other hand, being older is negatively associated with social media usage time. Intuitively, this result is in alignment with the common notion that the older generations are not as familiar with or interested in digital services compared to younger people. However, it can also be viewed through the function of social comparison through social media [27,56]. Specifically, as the Social Comparison Theory suggests, when lacking objective measures, people tend to compare themselves to others to gain a sense of self-evaluation [57]. Younger people relatively lack concrete self-established values compared to the older age groups with more life experiences. In this case, social media is a platform to gain other forms of perceived social validation such as the number of followers, likes, or comments [27].

Lastly, higher income means more options for social interactions and entertainment. Particularly in the context of Qatar, there exists a wide range of premium entertainment services and social events for financially abundant individuals. The capability to afford these options likely decreases the desire and time available for spending on social media.

The analysis results show that among those with high PSR degrees with their favorite Instagram influencer(s), females, young people, and poor people tend to use social media for more hours each day. These findings confirmed the directions of found

patterns in the demographic factors upon interacting with PSR toward social media usage time. There are a couple of noteworthy aspects that can be further considered, regarding the interactions of factors and the regional context of the studied population. Although women's empowerment in Qatar has been on the rise in the past few decades, traditional social norms still hold certain biased views against women [58,59]. Thus, it is possible that female users may develop a stronger attachment to idolized online figures or influencers that provide a sense of social security and comfort. Regarding PSRs among younger social media users, the formation of tightly-knit fandom communities or subcultures, where members often use internet slang and unique communication patterns extensively [60] can increase the appeal toward in-group engagement. This may reinforce the sense of belongingness and commitment, and thus being associated with greater usage time. Regarding financial capability, individuals having strong PSRs are more likely to engage in behaviors such as contributing to the fan economy financially [61,62]. For those with lower income, spending time on social media and engaging in financially affordable behaviors such as retweeting influencers' tweets [63] or watching them live-streaming [64] are ways to trade time for a sense of contribution to the PSRs.

## 5. Implications

The present study suggests that cultural and societal norms can have a considerable background influence in shaping the dynamics of social media usage behavior, which is in alignment with extant studies [27,65,66]. From a practical standpoint, the results suggest that interventions aimed at reducing excessive social media use should be tailored to the specific demographic and cultural context of the target population. In the case of Qatar, addressing the unique local characteristics of women, young generations, or lower-income groups might help increase effectiveness.

To be more specific, given that females with strong PSRs tend to use social media more extensively, policy makers could feature relatable female role models or influencers to promote balanced online–offline lifestyles. Such campaigns can speak directly to women's experiences, showing positive ways of engaging with influencers (e.g., seeking inspiration without excessive scrolling) while acknowledging underlying social pressures.

Also, because the findings suggest that younger users are especially prone to both higher PSRs and higher social media usage, programs that teach digital literacy and self-regulation (e.g., how to set healthy screen-time boundaries) could be integrated into high school or university curricula. These initiatives can reduce the risks of problematic use by encouraging purposeful engagement that still supports healthy identity exploration and peer bonding.

Lastly, for lower-income individuals who may rely heavily on cost-free entertainment such as social media, policymakers and local organizations could facilitate affordable offline social activities or community events. For example, subsidized access to sports clubs, public libraries, or cultural festivals can offer enjoyable and meaningful non-digital outlets.

Future work could explore how influencer-specific characteristics—such as gender, genre, or celebrity status—shape parasocial relationships (PSRs) and social media usage. Longitudinal and qualitative approaches may further deepen our understanding of how user–influencer dynamics evolve over time, especially in non-WEIRD contexts where distinct cultural factors play a key role.

## 6. Limitations

This study has some limitations. The data used for analysis has high skewness in some examined parameters due to the nature of digital social media usage (such as young age). However, the employed method of MCMC-aided Bayesian analysis helped increase inference accuracy when dealing with such skewed data. Data was also collected from users who followed Instagram influencers, thus may not fully represent patterns of PSRs in other platforms. Furthermore, participants were from Qatar, which might have different psychological nuances compared to social media users in other regions of the world. Additionally, given the generic nature of the original data regarding the PSRs, the demographic factors, and the influencers' characteristics, cautious approaches are recommended when exploring deeper into the matters based on the present study's results. Future studies should compare and update the patterns using data from people on other platforms and regions. Qualitative research is also particularly helpful in exploring further the issues behind the relationship between social media use and PSRs.

## Appendices

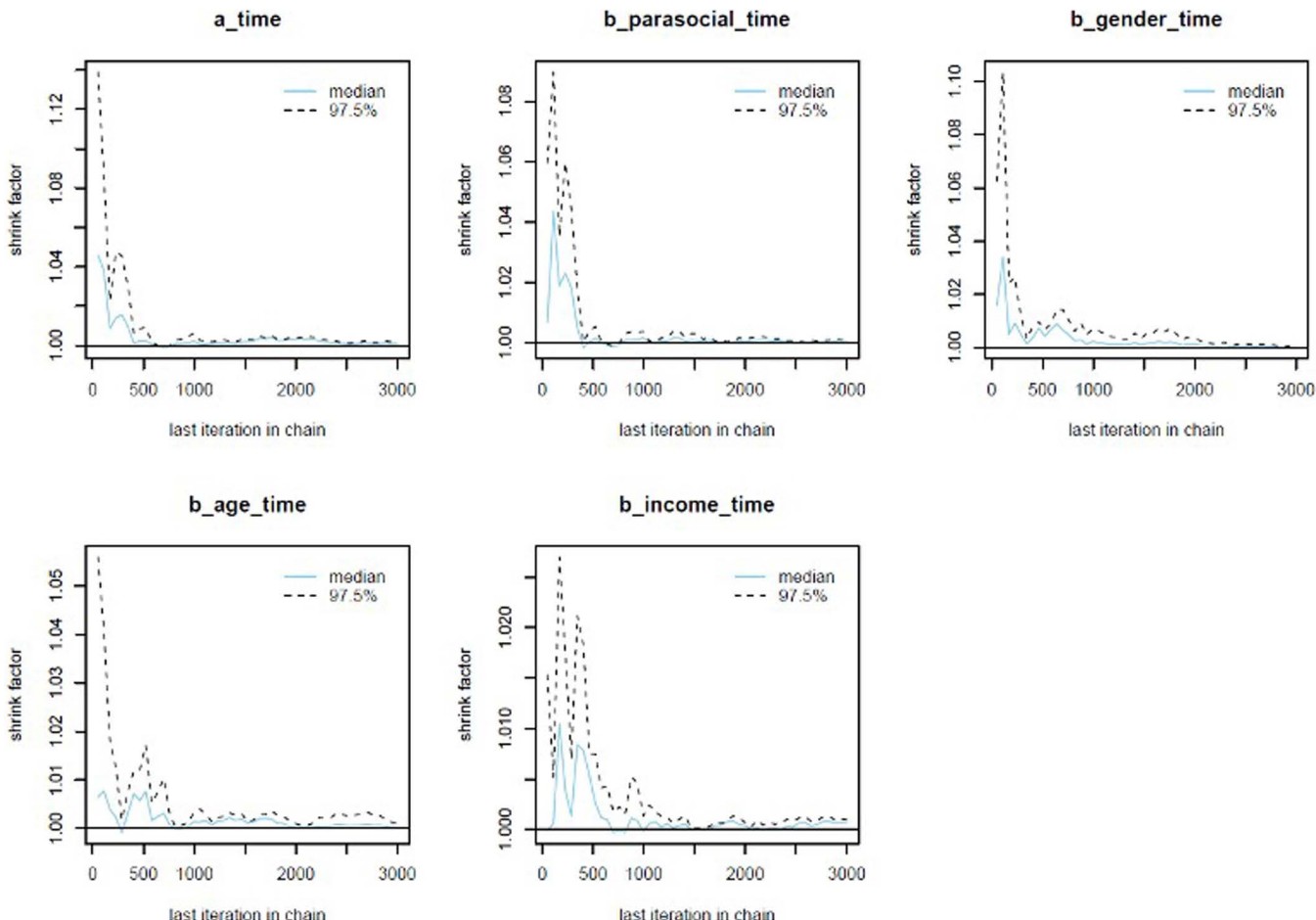

**Fig A1. Model 1's Gelman-Rubin-Brooks plots.**

**Fig A2. Model 1's autocorrelation plots.**

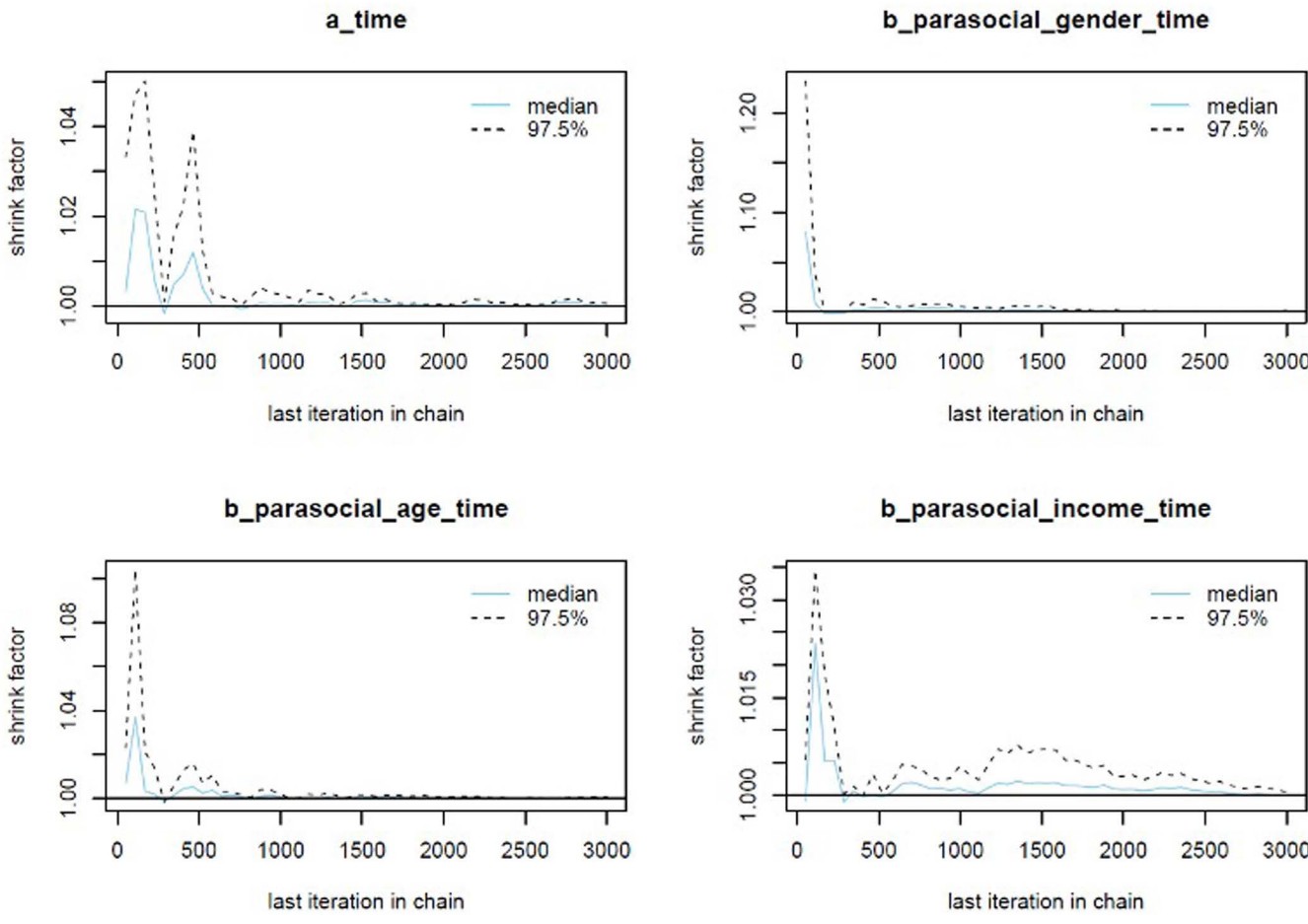

**Fig A3. Model 2's Gelman-Rubin-Brooks plots.**

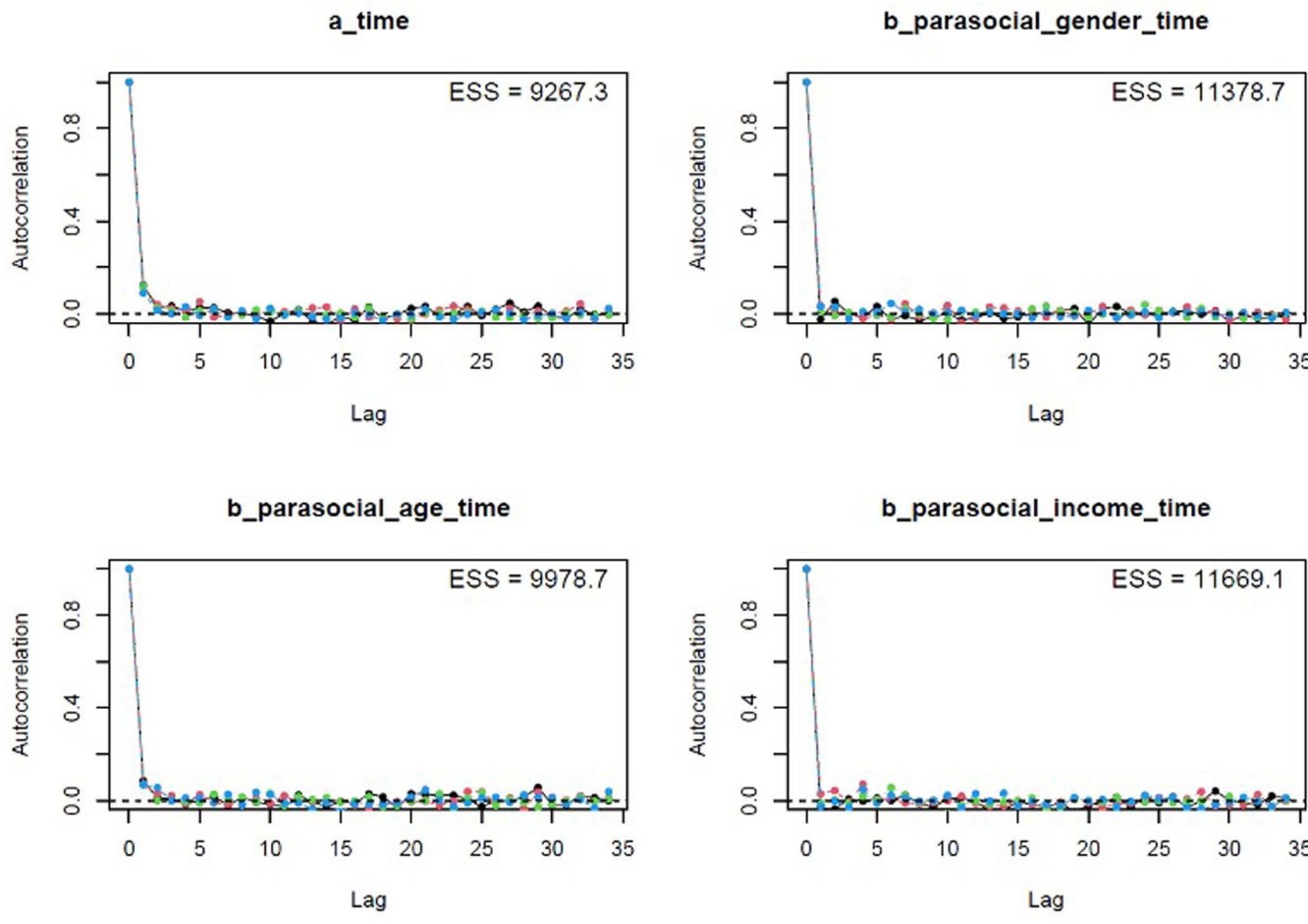

**Fig A4. Model 2's autocorrelation plots.**

## Acknowledgments

This paper is respectfully dedicated to Liu Tie, a cherished friend of the authors, who departed this life on December 26, 2023. His steadfast encouragement and support were invaluable. The authors are profoundly grateful for his enduring contributions. His loyal friendship remains a guiding force in their endeavors. This work serves to honor his enduring impact.

## Author contributions

**Conceptualization:** Ruining Jin, Tam-Tri Le.

**Investigation:** Ruining Jin.

**Methodology:** Tam-Tri Le.

**Supervision:** Tam-Tri Le.

**Visualization:** Tam-Tri Le.

**Writing – original draft:** Ruining Jin.

**Writing – review & editing:** Ruining Jin, Tam-Tri Le.

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
