## [Decision Letter · Decision Letter 0]

3 Dec 2024

PONE-D-24-40673Attachment Beyond the Screen: The Influences of Demographic Factors and Parasocial Relationships on Social Media Use in QatarPLOS ONE

Dear Dr. Jin,

Thank you for submitting your manuscript to PLOS ONE. After careful consideration, we feel that it has merit but does not fully meet PLOS ONE’s publication criteria as it currently stands. Therefore, we invite you to submit a revised version of the manuscript that addresses the points raised during the review process. Please take particular care to address the reviewers comments related to the rationale for your study and its contextualization in the wider literature.

We look forward to receiving your revised manuscript.

Kind regards,

Jennifer Tucker, PhD

Staff Editor

PLOS ONE

Journal Requirements: When submitting your revision, we need you to address these additional requirements. 1. Please ensure that your manuscript meets PLOS ONE's style requirements, including those for file naming. The PLOS ONE style templates can be found at https://journals.plos.org/plosone/s/file?id=wjVg/PLOSOne_formatting_sample_main_body.pdf and https://journals.plos.org/plosone/s/file?id=ba62/PLOSOne_formatting_sample_title_authors_affiliations.pdf 2. We notice that your supplementary figures are uploaded with the file type 'Figure'. Please amend the file type to 'Supporting Information'. Please ensure that each Supporting Information file has a legend listed in the manuscript after the references list.

Reviewers' comments:

Reviewer's Responses to Questions

**Comments to the Author**

1. Is the manuscript technically sound, and do the data support the conclusions?

Reviewer #1: Partly

Reviewer #2: Yes

2. Has the statistical analysis been performed appropriately and rigorously? 

Reviewer #1: I Don't Know

Reviewer #2: Yes

3. Have the authors made all data underlying the findings in their manuscript fully available?

Reviewer #1: Yes

Reviewer #2: Yes

4. Is the manuscript presented in an intelligible fashion and written in standard English?

Reviewer #1: Yes

Reviewer #2: Yes

5. Review Comments to the Author

Reviewer #1: Here are my thoughts on the manuscript for the authors' consideration:

Abstract

Since the study emphasizes the use of a “non-WEIRD” population, it may be beneficial to clarify this distinction within the Abstract's “Background” section. Briefly contextualizing why the non-WEIRD focus is essential would enhance the study’s framing.

It’s crucial, however, to avoid overstating the lack of research on problematic social media use and parasocial relationships (PSRs) in non-WEIRD populations, as this area has been explored. Specificity would improve accuracy, such as mentioning “non-WEIRD populations in the Middle East.” It’s important to avoid exaggerated claims; studies exist on social media use and PSRs in contexts like Iran:

https://link.springer.com/article/10.1007/s12671-023-02271-9

https://link.springer.com/article/10.1007/s40519-024-01658-4

Additionally, in the “Methods”, the phrase “participant followed Instagram influencers” requires more precision: what types of influencers were included, and how many influencers did participants need to follow to qualify? This would help clarify inclusion criteria.

In the Abstract's “Conclusions” section, the second and third sentences seem to extend beyond the study’s findings. Staying closely aligned with the study's results here would strengthen the rigor. Similarly, the first sentence in this section could be clarified to align with the study's goal of understanding social media usage time.

Introduction

You mentioned a few studies conducted on non-WEIRD populations in the Introduction (e.g., Iran). Since the “Current Study” section later underscores the scarcity of non-WEIRD research, it may be more coherent to focus on WEIRD literature until that section. Here, you can acknowledge non-WEIRD studies and note any research gaps.

Further arguments could clarify why analyzing WEIRD and non-WEIRD populations separately is conceptually meaningful. This may include discussing why findings from WEIRD samples may not directly generalize to non-WEIRD populations for phenomena like PSRs.

In the “Methodology,” be clear on what constitutes “participants who followed Instagram influencers.” For example, specify if following at least one influencer (e.g., a fitness or food influencer) on Instagram was required.

Also, it’s worth noting that your measure assesses time spent on social media, which differs from problematic or addictive social media use. The literature should reflect this distinction by discussing excessive social media time and its negative outcomes rather than problematic use.

For PSR measurement, since the study uses a tool assessing PSR with a single media figure (e.g., “I feel 000 is fascinating on his/her SNS”), it’s assessing PSR with a favorite influencer rather than a general set of influencers.

Discussion

In the Discussion and throughout the manuscript, clarity is needed. For instance, rather than saying, “The finding that a higher PSR degree is associated with higher daily social media usage time…,” specify “higher PSR with a favorite Instagram influencer.”

Given the generic nature of data regarding the PSR with a favorite influencer (e.g., gender, popularity, and expertise of the influencer, which may affect outcomes), a more cautious interpretation in the discussion and conclusions is warranted.

I hope these suggestions are helpful.

Reviewer #2: 1. In section 1.1.Excessive social media use and its outcomes:It would be better to include statistics on Non-WEIRD countries if it is possible.

2. Is there any research that has conducted on your research variables in Non-WEIRD countries that can support your findings? If there is, you can use it in the discussion section.

3. The authors of this article are expected to explain that why the data collection took place in 2020 and the article was written about 4 years later.

4. It is suggested that additional editing be done on the paper to meet the criteria of standard English language.

6. PLOS authors have the option to publish the peer review history of their article (what does this mean? ). If published, this will include your full peer review and any attached files.

**Do you want your identity to be public for this peer review?** For information about this choice, including consent withdrawal, please see our Privacy Policy .

Reviewer #1: No

Reviewer #2: No

---

## [Author Response · Author response to Decision Letter 1]

9 Dec 2024

Plos One

December 6, 2024

Dear Editors and the Editorial Office:

Submission of a revised manuscript

Thank you very much for spending a great amount of time and effort reviewing our manuscript. We would like to submit a revised manuscript titled “Attachment Beyond the Screen: The Influences of Demographic Factors and Parasocial Relationships on Social Media Use in Qatar”.

In the revised version, we have made the following major changes:

• Abstract has been adjusted based on suggestions

• Introduction and literature review have gone through a major rework

• Methods have been articulated

• Discussion has been enhanced to reflect academic rigor and be aligned with literature discussed in the introduction

• Limitations have been expanded

• Wording and language have been strengthened

In our following responses, we have integrated the issues addressed by the editor together with similar concerns raised by the reviewers. We have addressed point-to-point responses to the comments of reviewers in our updated version. Please notice that in the revised paper, the parts that are highlighted in yellow are our revisions based on your feedback. Below are our modifications and answers (emboldened) to Reviewer 1 and Reviewer 2 (in italics).

First of all, we would like to express our sincere appreciation for the reviewers’ professionalism, goodwill, and meticulous review report. This is a warm feeling considering the current state of the modern academic publishing system. With their help, we have improved the manuscript significantly in many aspects. We present our responses to their detailed suggestions below.

Reviewer 1

Abstract

Since the study emphasizes the use of a “non-WEIRD” population, it may be beneficial to clarify this distinction within the Abstract's “Background” section. Briefly contextualizing why the non-WEIRD focus is essential would enhance the study’s framing.

We are deeply thankful for your insightful comments. We have now revised our abstract to reflect this point. Please see below:

Abstract

Background: Most studies on social media usage and parasocial relationships (PSRs) have been conducted in WEIRD (Western, Educated, Industrialized, Rich, and Democratic) societies, potentially overlooking the unique cultural, social, and economic factors present in non-WEIRD contexts. Examining these phenomena in a non-WEIRD setting is essential for a comprehensive understanding of social media's global impact.

Methods: Secondary data from 574 participants in Qatar who followed Instagram influencers were analyzed using Bayesian analyses aided by Markov Chain Monte Carlo (MCMC) algorithms to examine the relationships between social media usage time, PSRs, and demographic factors.

Findings: The analysis results show that, regarding linear effects, a stronger parasocial relationship with Instagram influencer(s) is associated with higher daily social media usage time. Meanwhile, being male, being older, and having higher incomes all have negative associations with daily social media usage time. When parasocial relationships and the three demographic factors are seen in their interactions, negative associations with social media usage were also found in a similar pattern. To elaborate, among those with high parasocial relationship degrees, females, young people, and poor people tend to use social media for more hours each day.

Conclusions: This study highlights that demographic factors such as gender, age, and income in their interactions with parasocial relationships can influence social media usage time within the non-WEIRD social context of Qatar. The findings underscore the necessity of considering the specific local cultural settings when studying social media behaviors.

It’s crucial, however, to avoid overstating the lack of research on problematic social media use and parasocial relationships (PSRs) in non-WEIRD populations, as this area has been explored. Specificity would improve accuracy, such as mentioning “non-WEIRD populations in the Middle East.” It’s important to avoid exaggerated claims; studies exist on social media use and PSRs in contexts like Iran:

https://link.springer.com/article/10.1007/s12671-023-02271-9

https://link.springer.com/article/10.1007/s40519-024-01658-4

Thank you very much for your suggestion and academic rigor. We have articulated the claims and revised the introduction accordingly. Please see below:

1.6. Current study

However, findings from WEIRD contexts may not directly generalize to non-WEIRD populations, because various cultural, socioeconomic, and social norms might significantly influence the social media use pattern, and the formation and impact of parasocial relationships. Currently, studies on social media use in non-WERID contexts offered complicated findings. For example, Chinese netizens average about 2 hours per day social media use time [26]. One study on Chinese adolescents suggested that social media use time alone is not associated with increased anxiety toward their body image. However, when seeking attention through the use of social media, the more social media use time Chinese adolescents, the greater they would suffer from social appearance anxiety [27]. On the other hand, India has more than 600 million social media users [28], one studies on Indian adolescents offered similar findings, suggesting that long time use of social media is associated with increased stress, anxiety, and depression [29]. However, few studies have been conducted on non-WEIRD populations in the Middle East. One cross-cultural study examined the mindful use of social media among Iranian and American users, finding that Iranian participants reported higher levels of mindful social media use, which were associated with lower symptoms of social media addiction [30]. Furthermore, one recent study investigated the role of parasocial relationships with favorite food influencers among Iranian social media users. Their study found that stronger PSRs with food influencers were associated with higher levels of eating disorder symptoms, food addiction, and grazing behaviors [31]. However, the aforementioned studies primarily examined social media use and parasocial relationships (PSRs) within Iran, leaving a significant gap in research on non-WEIRD populations in the broader Middle Eastern context. Qatar’s unique context offers a distinctive environment where rapid economic development and high internet penetration coexist with deeply rooted cultural and traditional norms. This blend creates a unique dynamic in how individuals engage with social media and form PSRs with influencers.

[…]

Additionally, in the “Methods”, the phrase “participant followed Instagram influencers” requires more precision: what types of influencers were included, and how many influencers did participants need to follow to qualify? This would help clarify inclusion criteria.

Thank you for your insightful comment. We appreciate your suggestion to enhance the clarity and quality of the manuscript, and provide additional information based on the data paper information.

2.1. Materials and variables

We use secondary data from the data article “Instagram Influencers Attributes and Parasocial Relationship: A dataset from Qatar” [31]. Research ethics approval for the data collection by [31] was obtained from the Qatar University Review Board (number QU-IRB 1195-E/19). [31] declared that participation was entirely voluntary, all respondents were systematically informed about the study’s content and objectives before participation, and all respondents gave informed consent to participate. The data collection happened in 2020, from January 29th to February 16th.

The dataset contains survey information from 574 participants living in Qatar who followed Instagram influencers. Participants were required to follow at least one Instagram influencer from the following areas of expertise: Fashion (N=26, 4.5%), Traveling (N=30, 5.2%), Beauty Products (N=16, 2.8%), Food and Beverages (N=44, 7.7%), Others (N=272, 47.4%), and Multiple (N=186, 32.4%).

[…]

In the Abstract's “Conclusions” section, the second and third sentences seem to extend beyond the study’s findings. Staying closely aligned with the study's results here would strengthen the rigor. Similarly, the first sentence in this section could be clarified to align with the study's goal of understanding social media usage time.

Once again, we appreciate your academic rigor and made revisions accordingly in the “Conclusions” part of the abstract. Please see our answers above.

Introduction

You mentioned a few studies conducted on non-WEIRD populations in the Introduction (e.g., Iran). Since the “Current Study” section later underscores the scarcity of non-WEIRD research, it may be more coherent to focus on WEIRD literature until that section. Here, you can acknowledge non-WEIRD studies and note any research gaps.

Further arguments could clarify why analyzing WEIRD and non-WEIRD populations separately is conceptually meaningful. This may include discussing why findings from WEIRD samples may not directly generalize to non-WEIRD populations for phenomena like PSRs.

Thank you! We have followed your suggestion to move studies on non-WEIRD contexts in the “current study” section. Please also see above for our response.

In the “Methodology,” be clear on what constitutes “participants who followed Instagram influencers.” For example, specify if following at least one influencer (e.g., a fitness or food influencer) on Instagram was required.

Thank you for your suggestion! We have included additional information regarding the expertise of the influencer on Instagram. Please see the revised Materials subsection in the above answer.

Also, it’s worth noting that your measure assesses time spent on social media, which differs from problematic or addictive social media use. The literature should reflect this distinction by discussing excessive social media time and its negative outcomes rather than problematic use.

Thank you for pointing this out. We have removed the discussion about problematic or addictive social media use, now focusing on excessive social media time and its negative outcomes.

For PSR measurement, since the study uses a tool assessing PSR with a single media figure (e.g., “I feel 000 is fascinating on his/her SNS”), it’s assessing PSR with a favorite influencer rather than a general set of influencers.

Discussion

In the Discussion and throughout the manuscript, clarity is needed. For instance, rather than saying, “The finding that a higher PSR degree is associated with higher daily social media usage time…,” specify “higher PSR with a favorite Instagram influencer.”

Given the generic nature of data regarding the PSR with a favorite influencer (e.g., gender, popularity, and expertise of the influencer, which may affect outcomes), a more cautious interpretation in the discussion and conclusions is warranted.

I hope these suggestions are helpful.

We appreciate this reminder for cautious wording. We have made revisions in the discussion and other places to improve clarity.

4. Discussion

The analysis results show that, regarding linear effects, a higher PSR with one or multiple favorite Instagram influencers is associated with higher daily social media usage time. Meanwhile, being male, being older, and having higher incomes all have negative associations with daily social media usage time. When PSRs and the three demographic factors are seen in their interactions, negative associations with social media usage were also found in a similar pattern. To elaborate, among those with high PSR degrees with their favorite Instagram influencer(s), females, young people, and poor people tend to use social media for more hours each day.

The finding that a higher PSR degree with Instagram influencer(s) is associated with higher daily social media usage time is in alignment with prior studies [22, 23]. Such a finding can be interpreted through a two-way influence. Firstly, when users have a strong PSR with influencers, they are naturally more inclined to engage with the influencers' content (viewing, commenting, liking, sharing, etc.), which directs the social media platforms to automatically refine the provision of their desired content through algorithms. Thus, the platforms deliver more tailored content that aligns with the user's interests, particularly content related to the influencers they follow [44, 45]. This personalized content loop would externally contribute to increased social media usage, as users are drawn into a continuous cycle of engagement [26]. Reversely, as users spend more time on social media, there is a higher chance for them to come across influencers’ content that is engaging and stimulating, which helps form or reinforce the perceived connections between followers and influencers [46]. The felt attachment due to various psychological factors/reasons, as suggested in the UGT [24, 25], can cause a PSR to be strengthened by this loop mechanism of social media content engagement.

The study findings that being male, being older, and having higher incomes have a negative association with daily social media usage time are consistent with earlier studies [14–16, 47–49]. Here, we can take into account the specific cultural perspective of the studied population. One possible aspect is that when males in Qatar use social media, they also consider the socially expected masculine characteristics, as well as traditional masculine roles in the family and society, such as showcasing socioeconomic status [50], or highlighting the provider-income maker identity in the family [51, 52]. In this sense, long social media usage time might be perceived as excessive leisure or gossiping, undermining male users’ ideal image (both self-image and social image). Thus, males’ social media usage time is negatively associated with their gender identity.

On the other hand, being older is negatively associated with social media usage time. Intuitively, this result is in alignment with the common notion that the older generations are not as familiar with or interested in digital services compared to younger people. However, it can also be viewed through the function of social comparison through social media [26, 53]. Specifically, as the Social Comparison Theory suggests, when lacking objective measures, people tend to compare themselves to others to gain a sense of self-evaluation [54]. Younger people tend to relatively lack concrete self-established values compared to the older age groups with more life experiences. In this case, social media is a platform to gain other forms of perceived social validation such as the number of followers, likes, or comments [26].

Lastly, higher income means more options for social interactions and entertainment. Particularly in the context of Qatar, there exists a wide range of premium entertainment services and social events for financially abundant individuals. The capability to afford these options likely decreases the desire and time available for spending on social media.

The analysis results show that among those with high PSR degrees with their favorite Instagram influencer(s), females, young people, and poor people tend to use social media for more hours each day. These findings confirmed the directions of found patterns in the demographic factors upon interacting with PSR toward social media usage time. There are a couple of noteworthy aspects that can be further considered, regarding the interactions of factors and the regional context of the studied population. Although women’s empowerment in Qatar has been on the rise in the past few decades, traditional social norms still hold certain biased views against women [55, 56]. Thus, it is possible that female users may develop a stronger attachment to idolized online figures or influencers that provide a sense of social security and comfort. Regarding PSRs among younger social media users, the formation of tightly-knit fandom communities or subcultures, where members often use internet slang and unique communication patterns extensively [57] can increase the appeal toward in-group engagement. This may reinforce the sense of belongingness and commitment, and thus extend usage time. Regarding financ

---

## [Decision Letter · Decision Letter 1]

21 Apr 2025

PONE-D-24-40673R1Attachment Beyond the Screen: The Influences of

Demographic Factors and Parasocial Relationships on Social Media Use in QatarPLOS ONE

Dear Dr. Jin,

Thank you for submitting your manuscript to PLOS ONE. After careful consideration, we feel that it has merit but does not fully meet PLOS ONE’s publication criteria as it currently stands. Therefore, we invite you to submit a revised version of the manuscript that addresses the points raised during the review process.

We look forward to receiving your revised manuscript.

Kind regards,

Andrea Fronzetti Colladon, Ph.D.

Academic Editor

PLOS ONE

Reviewers' comments:

Reviewer's Responses to Questions

**Comments to the Author**

1. If the authors have adequately addressed your comments raised in a previous round of review and you feel that this manuscript is now acceptable for publication, you may indicate that here to bypass the “Comments to the Author” section, enter your conflict of interest statement in the “Confidential to Editor” section, and submit your "Accept" recommendation.

Reviewer #2: All comments have been addressed

Reviewer #3: All comments have been addressed

2. Is the manuscript technically sound, and do the data support the conclusions?

Reviewer #2: Yes

Reviewer #3: Yes

3. Has the statistical analysis been performed appropriately and rigorously? 

Reviewer #2: Yes

Reviewer #3: Yes

4. Have the authors made all data underlying the findings in their manuscript fully available?

Reviewer #2: Yes

Reviewer #3: Yes

5. Is the manuscript presented in an intelligible fashion and written in standard English?

Reviewer #2: Yes

Reviewer #3: Yes

6. Review Comments to the Author

Reviewer #2: Dear authors

The paper is generally well written and structured and all comments have been addressed.

Regards,

Reviewer #3: Thank you for the opportunity to review the revised manuscript. This study presents a valuable contribution to the literature on social media behavior by focusing on a non-WEIRD population in Qatar; an underexplored yet important context. The manuscript is well-written, and the authors have clearly made substantial revisions in response to earlier feedback. That said, several areas still require clarification or refinement. Here is my detailed feedback:

1. The manuscript mentions use of a six-item PSR scale but does not present the items or explain how they were adapted for this cultural context. Given the centrality of this construct, please consider including the items in an appendix or table.

2. At various points (e.g., “PSRs cause increased social media use”), causal implications are made. Please rephrase to emphasize correlational interpretation throughout. This is particularly important given the cross-sectional nature of the data.

3. The dataset was collected in early 2020. Although the authors now mention this and explain their rationale for use, I recommend briefly reflecting on how Instagram use or influencer engagement in Qatar may have evolved since that time—particularly given the impact of COVID-19 on digital behavior.

4. The Discussion is well developed, however, some parts of it just repeat earlier findings without sufficient interpretation. I suggest revising the Discussion further to focus more on implications and conceptual takeaways rather than just repeating the results.

5. In the future research directions, some further limitations should be addressed. For instance, the inclusion of influencer characteristics (e.g., gender, genre, celebrity status) could be explored in future PSR studies. Also, the potential for longitudinal or qualitative research to deepen understanding of user-influencer dynamics should be highlighted here.

6. Line 116: “usa time” should be corrected to “usage time.”

7. Consider adding a brief explanation of why MCMC was preferred over frequentist approaches for readers unfamiliar with Bayesian methods.

I hope the authors find my comments helpful in improving the work further.

7. PLOS authors have the option to publish the peer review history of their article (what does this mean? ). If published, this will include your full peer review and any attached files.

**Do you want your identity to be public for this peer review?** For information about this choice, including consent withdrawal, please see our Privacy Policy .

Reviewer #2: No

Reviewer #3: No

---

## [Author Response · Author response to Decision Letter 2]

22 Apr 2025

Plos One

April 22, 2025

Dear Editors and the Editorial Office:

Submission of a revised manuscript

Thank you very much for spending a great amount of time and effort reviewing our manuscript. We would like to submit a revised manuscript titled “Attachment Beyond the Screen: The Influences of Demographic Factors and Parasocial Relationships on Social Media Use in Qatar”.

In the revised version, we have made the following major changes:

1. Intro & Methods section have been expanded to enhance academic rigor

2. Discussions have been enhanced to add additional depth and clarity

3. Implications have been rewritten to offer more substantial suggestions

In our following responses, we have integrated the issues addressed by the editor together with similar concerns raised by the reviewers. We have addressed point-to-point responses to the comments of reviewers in our updated version. Please notice that in the revised paper, the parts that are highlighted in yellow are our revisions based on your feedback. Below are our modifications and answers (emboldened) to Reviewer 2, and Reviewer 3 (in italics).

First of all, we would like to express our sincere appreciation for the reviewers’ professionalism, goodwill, and meticulous review report. This is a warm feeling considering the current state of the modern academic publishing system. With their help, we have improved the manuscript significantly in many aspects. We present our responses to their detailed suggestions below.

Reviewer 2

The paper is generally well written and structured and all comments have been addressed.

Thank you very much for your kind and encouraging feedback. We truly appreciate your time and thoughtful review. Your comments have been invaluable in helping us improve the overall quality of the manuscript—thank you again for your support.

Reviewer 3

Thank you for the opportunity to review the revised manuscript. This study presents a valuable contribution to the literature on social media behavior by focusing on a non-WEIRD population in Qatar; an underexplored yet important context. The manuscript is well-written, and the authors have clearly made substantial revisions in response to earlier feedback.

Thank you for your kind words.

That said, several areas still require clarification or refinement. Here is my detailed feedback:

1. The manuscript mentions use of a six-item PSR scale but does not present the items or explain how they were adapted for this cultural context. Given the centrality of this construct, please consider including the items in an appendix or table.

Thank you for your academic rigor. We have added the Questionnaire at the end of the manuscript, highlighting the items used in this study to measure PSR. Please see below:

Questionnaire in English

Part A: General Information

1. How many hours do you spend on social media every day?

Never � 1-2 hours � 3-4 hours � 5 hours or above

2. Which Instagram influencer do you mostly follow? (mention one influencer)

--------------------------------

3. When did you start following this Instagram influencer?

less than 6 months ago

One year ago

Two years ago

Three years ago or more

4. What is the area of expertise of this Instagram influencer?

Fashion

Traveling

Beauty products

Food and beverages

Others: --------------------------------

Part B: Rating Statements

To what extent do you agree on the following statements?

(1) Strongly disagree, (2) Disagree (3) Neutral (4) Agree (5) Strongly agree

Statement 1 2 3 4 5

Hom1: This Instagram influencer thinks like me.

Hom2: This Instagram influencer is similar to me.

Hom3: This Instagram influencer is like me.

Hom4: This Instagram influencer shares my values.

Hom5: This Instagram influencer has a lot in common with me.

Hom6: Instagram influencer behaves like me.

Hom7: This Instagram influencer has thoughts and ideas that are similar to mine.

Hom8: I think that my Instagram influencer could be a friend of mine.

Hom9: I would like to have a friendly chat with my Instagram influencer.

Pop1: This Instagram influencer has a high exposure in the Instagram environment.

Pop2: This Instagram influencer has a high popularity in the Instagram environment.

Pop3: This Instagram influencer has a high reputation in the Instagram environment.

Lev1: This Instagram influencer can cause debate in the Instagram environment.

Lev2: This Instagram influencer is topical in the Instagram environment.

Lev3: This Instagram influencer remarks in the Instagram environment are sensational.

Fash1: This Instagram influencer can lead the trend in the Instagram environment.

Fash2: This Instagram influencer is very fashionable.

Fash3: This Instagram influencer is very sensitive to fashion.

Aff1: This Instagram influencer is very close to people.

Aff2: This Instagram influencer behaviour is in a popular style.

Aff3: This Instagram influencer is a very down-to-earth person.

PSI1: I feel close enough to my favourite Instagram influencer to use his(her) Instagram.

PSI2: I feel comfortable about my favourite Instagram influencer messages.

PSI3: I can rely on information I get from my favourite Instagram influencer.

PSI4: I feel fascinated with my favourite Instagram influencer’s Instagram.

PSI5: In the past, I pitied my favourite Instagram influencer when he/she made a mistake on his/her Instagram.

PSI6: I think that my favourite Instagram influencer’s Instagram is helpful for my interests (in fashion and others).

WOM1: I am likely to say positive things about what my Instagram influencer promotes to others.

WOM2: I would recommend what my Instagram influencer promotes to my friends and relatives.

WOM3: If my friends were looking for a product or service of this type, would recommend what my Instagram influencer said about it.

Int1: I will buy the product or the service that Instagram influencer promoted through Instagram.

Int2: I have the intention to buy the product or the service that my Instagram influencer promoted on Instagram.

Int3: I am interested in buying the product or the service my Instagram influencer promoted on Instagram.

Int4: It is likely that I will buy the products or services my Instagram influencer promoted on Instagram in the future.

Int5: Overall, I am pleased with what my Instagram influencer promotes on Instagram.

Part C: Demographic Information

1. What is your age?

18 - 24 years old �25 - 34 years old � 35 - 44 years old

45 - 54 years old �55 - 64 years old � 64 years & above

2. What is your gender?

Male � Female

3. What is your nationality?

Qatari �Non-Qatari

6. What is your annual income?

Less than 50,000 Qatari Riyals

50,001-150,000 Qatari Riyals

150,001-250,000 Qatari Riyals

250,001-350,000 Qatari Riyals

350,001-450,000 Qatari Riyals

450,001 Qatari Riyals or above

2. At various points (e.g., “PSRs cause increased social media use”), causal implications are made. Please rephrase to emphasize correlational interpretation throughout. This is particularly important given the cross-sectional nature of the data.

Yes, you are right, and we are thankful for your insightful comments. We have adjusted our wording to address your concerns.

…

Conclusions: This study highlights that demographic factors such as gender, age, and income in their interactions with parasocial relationships are associated with social media usage time within the non-WEIRD social context of Qatar. The findings underscore the necessity of considering the specific local cultural settings when studying social media behaviors.

…

In the case of social media users, susceptible demographic groups may desire to gratify various social identity-based needs through interactions with influencers and celebrities, which might be associated with extensive social media usage time and PSRs.

…

This personalized content loop would be associated with increased social media usage, as users are drawn into a continuous cycle of engagement [27]. Reversely, as users spend more time on social media, there is a higher chance for them to come across influencers’ content that is engaging and stimulating, which helps form or reinforce the perceived connections between followers and influencers [49]. The felt attachment due to various psychological factors/reasons, as suggested in the UGT [24, 25], can further strengthen a PSR through this reciprocal loop of social media content engagement.

…

Regarding PSRs among younger social media users, the formation of tightly-knit fandom communities or subcultures, where members often use internet slang and unique communication patterns extensively [60] can increase the appeal toward in-group engagement. This may reinforce the sense of belongingness and commitment, and thus being associated with greater usage time.

…

3. The dataset was collected in early 2020. Although the authors now mention this and explain their rationale for use, I recommend briefly reflecting on how Instagram use or influencer engagement in Qatar may have evolved since that time—particularly given the impact of COVID-19 on digital behavior.

Thank you for your suggestion! We have added additional information regarding Instagram development in Qatar. Please see below:

…

In fact, recent estimates suggest that Instagram’s penetration in Qatar grew by over 10% between 2020 and 2022, and influencers took on a more prominent role in disseminating public information and shaping consumer habits during the pandemic [4,5]. As of January 2024, Instagram users have been reported to reach 1.7 million in 2024 (up from 1.1 million in 2023) with 35.1% female users and 64.9% male users[33].

As a result of the limited research on non WEIRD Middle Eastern populations and the rapid, COVID 19–induced shifts in Instagram adoption and influencer engagement, understanding these dynamics in Qatar’s digital spaces not only contributes to the social media development in Qatar but also offers novel insights into similar regions of quickly developing or emerging economies, where technological development and modernization meet with conservative values and ideologies.

…

4. The Discussion is well developed, however, some parts of it just repeat earlier findings without sufficient interpretation. I suggest revising the Discussion further to focus more on implications and conceptual takeaways rather than just repeating the results.

5. In the future research directions, some further limitations should be addressed. For instance, the inclusion of influencer characteristics (e.g., gender, genre, celebrity status) could be explored in future PSR studies. Also, the potential for longitudinal or qualitative research to deepen understanding of user-influencer dynamics should be highlighted here.

We thank you for your thoughtful feedback. Now we have made revisions in the implications and future studies section:

Implications

The present study suggests that cultural and societal norms can have a considerable background influence in shaping the dynamics of social media usage behavior, which is in alignment with extant studies [27, 65, 66]. From a practical standpoint, the results suggest that interventions aimed at reducing excessive social media use should be tailored to the specific demographic and cultural context of the target population. In the case of Qatar, addressing the unique local characteristics of women, young generations, or lower-income groups might help increase effectiveness.

To be more specific, given that females with strong PSRs tend to use social media more extensively, policy makers could feature relatable female role models or influencers to promote balanced online–offline lifestyles. Such campaigns can speak directly to women’s experiences, showing positive ways of engaging with influencers (e.g., seeking inspiration without excessive scrolling) while acknowledging underlying social pressures.

Also, because the findings suggest that younger users are especially prone to both higher PSRs and higher social media usage, programs that teach digital literacy and self-regulation (e.g., how to set healthy screen-time boundaries) could be integrated into high school or university curricula. These initiatives can reduce the risks of problematic use by encouraging purposeful engagement that still supports healthy identity exploration and peer bonding.

Lastly, for lower-income individuals who may rely heavily on cost-free entertainment such as social media, policymakers and local organizations could facilitate affordable offline social activities or community events. For example, subsidized access to sports clubs, public libraries, or cultural festivals can offer enjoyable and meaningful non-digital outlets.

Future work could explore how influencer-specific characteristics—such as gender, genre, or celebrity status—shape parasocial relationships (PSRs) and social media usage. Longitudinal and qualitative approaches may further deepen our understanding of how user–influencer dynamics evolve over time, especially in non-WEIRD contexts where distinct cultural factors play a key role.

6. Line 116: “usa time” should be corrected to “usage time.”

We are sorry for this typo and the issue has been fixed.

7. Consider adding a brief explanation of why MCMC was preferred over frequentist approaches for readers unfamiliar with Bayesian methods..

Yes, this is indeed thoughtful to enhance the manuscript’s readability. We have added additional explanations why MCMC was preferred.

…

the high skewness in demographic factors (due to the nature of digital social media use) can negatively affect inference accuracy because of the low data points available in some categories. For example, because of the higher proportion of women (61.7% vs. 38.3% men), traditional frequentist analyses can yield less stable parameter estimates under such imbalances or low data counts in specific subgroups. By contrast, a Bayesian framework aided by MCMC simulations allows for more flexible handling of skewed data, as it models parameters as probability distributions rather than fixed values. This approach “borrows strength” from more populated subgroups while explicitly accounting for uncertainty in underrepresented categories.

We sincerely thank the editor and reviewers for their thoughtful comments and valuable suggestions, which have substantially improved the clarity and depth of this manuscript. Their insights prompted us to expand our discussion on the implications of cultural and societal norms, refine our methodological justifications, and outline further avenues for investigation. We appreciate the time and effort they invested in reviewing our work, and we look forward to continuing this scholarly dialogue.

---

## [Decision Letter · Decision Letter 2]

4 June 2025

Attachment Beyond the Screen: The Influences of

Demographic Factors and Parasocial Relationships on Social Media Use in Qatar

PONE-D-24-40673R2

Dear Dr. Jin,

We’re pleased to inform you that your manuscript has been judged scientifically suitable for publication and will be formally accepted for publication once it meets all outstanding technical requirements.

Kind regards,

Andrea Fronzetti Colladon, Ph.D.

Academic Editor

PLOS ONE

Reviewers' comments:

Reviewer's Responses to Questions

**Comments to the Author**

1. If the authors have adequately addressed your comments raised in a previous round of review and you feel that this manuscript is now acceptable for publication, you may indicate that here to bypass the “Comments to the Author” section, enter your conflict of interest statement in the “Confidential to Editor” section, and submit your "Accept" recommendation.

Reviewer #3: All comments have been addressed

2. Is the manuscript technically sound, and do the data support the conclusions?

Reviewer #3: Yes

3. Has the statistical analysis been performed appropriately and rigorously? 

Reviewer #3: Yes

4. Have the authors made all data underlying the findings in their manuscript fully available?

Reviewer #3: Yes

5. Is the manuscript presented in an intelligible fashion and written in standard English?

Reviewer #3: Yes

6. Review Comments to the Author

Reviewer #3: Thank you for addressing my comments. I have no further suggestions. Best of luck with your research.

7. PLOS authors have the option to publish the peer review history of their article (what does this mean? ). If published, this will include your full peer review and any attached files.

**Do you want your identity to be public for this peer review?** For information about this choice, including consent withdrawal, please see our Privacy Policy .

Reviewer #3: No

---

## [Editor Report · Acceptance letter]

PONE-D-24-40673R2

PLOS ONE

Dear Dr. Jin,

I'm pleased to inform you that your manuscript has been deemed suitable for publication in PLOS ONE. Congratulations! Your manuscript is now being handed over to our production team.

Kind regards,

on behalf of

Prof. Andrea Fronzetti Colladon

Academic Editor

PLOS ONE